# Policy Learning for Fairness in Ranking

**Ashudeep Singh**
Department of Computer Science
Cornell University
Ithaca, NY 14850
ashudeep@cs.cornell.edu

**Thorsten Joachims**
Department of Computer Science
Cornell University
Ithaca, NY 14850
tj@cs.cornell.edu

## Abstract

Conventional Learning-to-Rank (LTR) methods optimize the utility of the rankings to the users, but they are oblivious to their impact on the ranked items. However, there has been a growing understanding that the latter is important to consider for a wide range of ranking applications (e.g. online marketplaces, job placement, admissions). To address this need, we propose a general LTR framework that can optimize a wide range of utility metrics (e.g. NDCG) while satisfying fairness of exposure constraints with respect to the items. This framework expands the class of learnable ranking functions to stochastic ranking policies, which provides a language for rigorously expressing fairness specifications. Furthermore, we provide a new LTR algorithm called FAIR-PG-RANK for directly searching the space of fair ranking policies via a policy-gradient approach. Beyond the theoretical evidence in deriving the framework and the algorithm, we provide empirical results on simulated and real-world datasets verifying the effectiveness of the approach in individual and group-fairness settings.

## 1 Introduction

Interfaces based on rankings are ubiquitous in today's multi-sided online economies (e.g., online marketplaces, job search, property renting, media streaming). In these systems, the items to be ranked are products, job candidates, or other entities that transfer economic benefit, and it is widely recognized that the position of an item in the ranking has a crucial influence on its exposure and economic success. Surprisingly, though, the algorithms used to learn these rankings are typically oblivious to the effect they have on the items. Instead, the learning algorithms blindly maximize the utility of the rankings to the users issuing queries to the systems [1], and there is evidence (e.g. [2, 3]) that this does not necessarily lead to rankings that would be considered fair or desirable.

In contrast to fairness in supervised learning for classification (e.g., [4–10]), fairness for rankings has been a relatively under-explored domain despite the growing influence of online information systems on our society and economy. In the work that does exist, some consider group fairness in rankings along the lines of demographic parity [11, 12], proposing definitions and methods that minimize the difference in the representation between groups in a prefix of the ranking [13–16]. Other recent works have argued that fairness of ranking systems corresponds to how they allocate exposure to individual items or group of items based on their merit [3, 17]. These works specify and enforce fairness constraints that explicitly link relevance to exposure in expectation or amortized over a set of queries. However, these works assume that the relevances of all items are known and they do not address the learning problem.

In this paper, we develop a Learning-to-Rank (LTR) algorithm – named FAIR-PG-RANK – that not only maximizes utility to the users, but that also rigorously enforces merit-based exposure constraints towards the items. Focusing on notions of fairness around the key scarce resource that search engines arbitrate, namely the relative allocation of exposure based on the items' merit, such fairness

constraints may be required to conform with anti-trust legislation [18], to alleviate winner-takes-all dynamics in a music streaming service [19], to implement anti-discrimination measures [20], or to implement some variant of search neutrality [21, 22]. By considering fairness already during learning, we find that FAIR-PG-RANK can identify biases in the representation that post-processing methods [3, 17] are, by design, unable to detect. Furthermore, we find that FAIR-PG-RANK performs better than heuristic approaches [23].

From a technical perspective, the main contributions of the paper are three-fold. First, we develop a conceptual framework in which it is possible to formulate fair LTR as a policy-learning problem subject to fairness constraints. We show that viewing fair LTR as learning a stochastic ranking policy leads to a rigorous formulation that can be addressed via Empirical Risk Minimization (ERM) on both the utility and the fairness constraint. Second, we propose a class of fairness constraints for ranking that incorporates notions of both individual and group fairness. And, third, we propose a policy-gradient method for implementing the ERM procedure that can directly optimize any information retrieval utility metric and a wide range of fairness criteria. Across a number of empirical evaluations, we find that the policy-gradient approach is a competitive LTR method in its own right, that FAIR-PG-RANK can identify and avoid biased features when trading-off utility for fairness, and that it can effectively optimize notions of individual and group fairness on real-world datasets.

## 2   Learning Fair Ranking Policies

The key goal of our work is to learn ranking policies where the allocation of exposure to items is not an accidental by-product of maximizing utility to the users, but where one can specify a merit-based exposure-allocation constraint that is enforced by the learning algorithm. An illustrative example adapted from Singh and Joachims [3] is that of ranking 10 job candidates, where the estimated probabilities of relevance (e.g., probability that an employer will invite for an interview) of 5 male job candidates are $\{0.89, 0.89, 0.89, 0.89, 0.89\}$ and those of 5 female candidates are $\{0.88, 0.88, 0.88, 0.88, 0.88\}$. Notice that these probabilities of relevance can themselves be gender-biased because of biased data or a biased prediction model. If these 10 candidates were ranked by these probabilities of relevance – thus maximizing utility to the users under virtually all information retrieval metrics [1] – the female candidates would get far less exposure (ranked 6,7,8,9,10) than the male candidates (ranked 1,2,3,4,5) even though they have almost the same estimated relevance. In this way, the ranking function itself is responsible for creating a strong *endogenous* bias against the female candidates, greatly amplifying and thus perpetuating any *exogenous* bias that may have led to small differences in the relevance estimates.

Addressing the endogenous bias created by the system itself, we argue that it should be possible to explicitly specify how exposure is allocated (e.g. make exposure proportional to relevance), that this specified exposure allocation is truthfully learned by the ranking policy (e.g. no systematic bias towards one of the groups), and that the ranking policy maintains a high utility to the users. Generalizing from this illustrative example, we develop our fair LTR framework as guided by the following three goals:

*Goal 1*:  Exposure allocated to an item is based on its merit. More merit means more exposure.
*Goal 2*:  Enable the explicit statement of how exposure is allocated relative to the merit of the items.
*Goal 3*:  Optimize the utility of the rankings to the users while satisfying *Goal 1* and *Goal 2*.

We will illustrate and further refine these goals as we develop our framework in the rest of this section. In particular, we first formulate the LTR problem in the context of empirical risk minimization (ERM) where exposure-allocation constraints are included in the empirical risk. We then define concrete families of allocation constraints for both individual and group fairness.

### 2.1   Learning to Rank as Policy Learning via ERM

Let $\mathcal{Q}$ be the distribution from which queries are drawn. Each query $q$ has a candidate set of documents $d^q = \{d_1^q, d_2^q, \dots d_{n(q)}^q\}$ that needs to be ranked, and a corresponding set of real-valued relevance judgments, $\mathrm{rel}^q = (\mathrm{rel}_1^q, \mathrm{rel}_2^q \dots \mathrm{rel}_{n(q)}^q)$. Our framework is agnostic to how relevance is defined, and it could be the probability that a user with query $q$ finds the document relevant, or it could be some subjective judgment of relevance as assigned by a relevance judge. Finally, each document $d_i^q$ is represented by a feature vector $x_i^q = \Psi(q, d_i^q)$ that describes the match between document $d_i^q$ and query $q$.

We consider stochastic ranking functions $\pi \in \Pi$, where $\pi(r|q)$ is a distribution over the rankings $r$ (i.e. permutations) of the candidate set. We refer to $\pi$ as a ranking policy and note that deterministic ranking functions are merely a special case. However, a key advantage of considering the full space of stochastic ranking policies is their ability to distribute expected exposure in a continuous fashion, which provides more fine-grained control and enables gradient-based optimization.

The conventional goal in LTR is to find a ranking policy $\pi^*$ that maximizes the expected utility of $\pi$

$$\pi^* = \operatorname{argmax}_{\pi \in \Pi} \mathbb{E}_{q \sim \mathcal{Q}}\big[U(\pi|q)\big],$$

where the utility of a stochastic policy $\pi$ for a query $q$ is defined as the expectation of a ranking metric $\Delta$ over $\pi$

$$U(\pi|q) = \mathbb{E}_{r \sim \pi(r|q)}\big[\Delta\big(r, \text{rel}^q\big)\big].$$

Common choices for $\Delta$ are DCG, NDCG, Average Rank, or ERR. For concreteness, we focus on NDCG as in [24], which is the normalized version of $\Delta_{\text{DCG}}(r, \text{rel}^q) = \sum_{j=1}^{n_q} \frac{u(r(j)|q)}{\log(1+j)}$, where $u(r(j)|q)$ is the utility of the document placed by ranking $r$ on position $j$ for $q$ as a function of relevance (e.g., $u(i|q) = 2^{\text{rel}_i^q} - 1$). NDCG normalizes DCG via $\Delta_{\text{NDCG}}(r, \text{rel}^q) = \frac{\Delta_{\text{DCG}}(r,\text{rel}^q)}{\max_r \Delta_{\text{DCG}}(r,\text{rel}^q)}$.

**Fair Ranking policies.** Instead of single-mindedly maximizing this utility measure like in conventional LTR algorithms, we include a constraint into the learning problem that enforces an application-dependent notion of fair allocation of exposure. To this effect, let's denote with $\mathcal{D}(\pi|q) \geq 0$ a measure of unfairness or the disparity, which we will define in detail in Section § 2.2. We can now formulate the objective of fair LTR by constraining the space of admissible ranking policies to those that have expected disparity less than some parameter $\delta$.

$$\pi_\delta^* = \operatorname{argmax}_\pi \mathbb{E}_{q \sim \mathcal{Q}}\left[U(\pi|q)\right] \text{ s.t. } \mathbb{E}_{q \sim \mathcal{Q}}\left[\mathcal{D}(\pi|q)\right] \leq \delta$$

Since we only observe samples from the query distribution $\mathcal{Q}$, we resort to the ERM principle and estimate the expectations with their empirical counterparts. Denoting the training set as $\mathcal{T} = \{(\mathbf{x}^q, \text{rel}^q)\}_{q=1}^N$, the empirical analog of the optimization problem becomes $\hat{\pi}_\delta^* = \operatorname{argmax}_\pi \frac{1}{N} \sum_{q=1}^N U(\pi|q)$ s.t. $\frac{1}{N} \sum_{q=1}^N \mathcal{D}(\pi|q) \leq \delta$. Using a Lagrange multiplier, this is equivalent to $\hat{\pi}_\delta^* = \operatorname{argmax}_\pi \min_{\lambda \geq 0} \frac{1}{N}\sum_{q=1}^N U(\pi|q) - \lambda\left(\frac{1}{N}\sum_{q=1}^N \mathcal{D}(\pi|q) - \delta\right)$. In the following, we avoid minimization w.r.t. $\lambda$ for a chosen $\delta$. Instead, we steer the utility/fairness trade-off by chosing a particular $\lambda$ and then computing the corresponding $\delta$ afterwards. This means we merely have to solve

$$\hat{\pi}_\lambda^* = \operatorname{argmax}_\pi \frac{1}{N} \sum_{q=1}^N U(\pi|q) - \lambda \frac{1}{N} \sum_{q=1}^N \mathcal{D}(\pi|q) \tag{1}$$

and then recover $\delta_\lambda = \frac{1}{N} \sum_{q=1}^N \mathcal{D}(\hat{\pi}_\lambda^*|q)$ afterwards. Note that this formulation implements our third goal from the opening paragraph, although we still lack a concrete definition of $\mathcal{D}$.

## 2.2 Defining a Class of Fairness Measures for Rankings

To make the training objective in Equation (1) fully specified, we still need a concrete definition of the unfairness measure $\mathcal{D}$. To this effect, we adapt the "Fairness of Exposure for Rankings" framework from Singh and Joachims [3], since it allows a wide range of application dependent notions of group-based fairness, including Statistical Parity, Disparate Exposure, and Disparate Impact. In order to formulate any specific disparity measure $\mathcal{D}$, we first need to define position bias and exposure.

**Position Bias.** The position bias of position $j$, $\mathbf{v}_j$, is defined as the fraction of users accessing a ranking who examine the item at position $j$. This captures how much attention an item will receive, where higher positions are expected to receive more attention than lower positions. In operational systems, position bias can be directly measured using eye-tracking [25], or indirectly estimated through swap experiments [26] or intervention harvesting [27, 28].

**Exposure.** For a given query $q$ and ranking distribution $\pi(r|q)$, the exposure of a document is defined as the expected attention that a document receives. This is equivalent to the expected position bias from all the positions that the document can be placed in. Exposure is denoted as $v_\pi(d_i)$ and can be expressed as

$$\text{Exposure}(d_i|\pi) = v_\pi(d_i) = \mathbb{E}_{r \sim \pi(r|q)}\big[\mathbf{v}_{r(d_i)}\big], \tag{2}$$

where $r(d_i)$ is the position of document $d_i$ under ranking $r$.

**Allocating exposure based on merit.** Our first two goals from the opening paragraph postulate that exposure should be based on an application dependent notion of merit. We define the *merit* of a document as a function of its relevance to the query (e.g., $\text{rel}_i$, $\text{rel}_i^2$ or $\sqrt{\text{rel}_i}$ depending on the application). Let's denote the merit of document $d_i$ as $M(\text{rel}_i) \geq 0$, or simply $M_i$, and we state that each document in the candidate set should get exposure proportional to its merit $M_i$.

$$\forall d_i \in d^q : \text{Exposure}(d_i|\pi) \propto M(\text{rel}_i)$$

For many queries, however, this set of exposure constraints is infeasible. As an example, consider a query where one document in the candidate set has relevance 1, while all other documents have small relevance $\epsilon$. For sufficiently small $\epsilon$, any ranking will provide too much exposure to the $\epsilon$-relevant documents, since we have to put these documents somewhere in the ranking. This violates the exposure constraint, and this shortcoming is also present in the Disparate Exposure measure of Singh and Joachims [3] and the Equity of Attention constraint of Biega et al. [17].

Note that this overabundance of exposure for some queries is not a fairness problem, since the extra exposure that some items receive does not come at the expense of other items. Furthermore, it is typically the items that have slightly lower merit that get disadvantaged by utility maximization, as illustrated in the introductory example. We thus replace the proportionality constraint with the following set of inequality constraints where $\forall d_i, d_j \in d^q$ with $M(\text{rel}_i) \geq M(\text{rel}_j) > 0$,

$$\frac{\text{Exposure}(d_i|\pi)}{M(\text{rel}_i)} \leq \frac{\text{Exposure}(d_j|\pi)}{M(\text{rel}_j)}$$

This one-sided set of constraints still enforce proportionality of exposure to merit, but allows the allocation of overabundant exposure which is achieved by only enforcing that higher-merit items don't get exposure beyond their merit. Note that the opposite direction of the constraint is already encouraged by utility maximization, where high-merit items tend to receive more exposure than they deserve.

Connecting this reasoning back to the example, after putting the item with relevance 1 at rank one, we have to put $\epsilon$-relevant items in position two and further. These $\epsilon$-relevant items are now overexposed which violates the two-sided constraint, but not the one-sided constraint. In this way, the one-sided metric together with utility maximization allows non-relevant items to get higher exposure when this is unavoidable in the tail of the ranking. In the other direction, the metric counteracts unmerited rich-get-richer dynamics, as present in the motivating example earlier.

**Measuring disparate exposure.** We can now define the following disparity measure $\mathcal{D}$ that captures in how far the fairness-of-exposure constraints are violated

$$\mathcal{D}_{\text{ind}}(\pi|q) = \frac{1}{|H_q|} \sum_{(i,j) \in H_q} \max\left[0, \frac{v_\pi(d_i)}{M_i} - \frac{v_\pi(d_j)}{M_j}\right], \tag{3}$$

where $H_q = \{(i, j) \text{ s.t. } M_i \geq M_j > 0\}$. The measure $\mathcal{D}_{\text{ind}}(\pi|q)$ is always non-negative and it equals zero only when the individual constraints are exactly satisfied.

**Group fairness disparity.** The disparity measure from above implements an individual notion of fairness, while other applications ask for a group-based notion. Here, fairness is aggregated over the members of each group. A group of documents can refer to sets of items sold by one seller in an online marketplace, to content published by one publisher, or to job candidates belonging to a protected group. Similar to the case of individual fairness, we want to allocate exposure to groups proportional to their merit. Hence, in the case of only two groups $G_0$ and $G_1$, we can define the following group fairness disparity for query $q$ as

$$\mathcal{D}_{\text{group}}(\pi|q) = \max\left(0, \frac{v_\pi(G_i)}{M_{G_i}} - \frac{v_\pi(G_j)}{M_{G_j}}\right), \tag{4}$$

where $G_i$ and $G_j$ are such that $M_{G_i} \geq M_{G_j}$ and $\text{Exposure}(G|\pi) = v_\pi(G) = \frac{1}{|G|} \sum_{d_i \in G} v_\pi(d_i)$ is the average exposure of group $G$, and the merit of the group $G$ is denoted by $M_G = \frac{1}{|G|} \sum_{d_i \in G} M_i$.

## 3  FAIR-PG-RANK: A Policy Learning Algorithm for Fair LTR

In the previous section, we defined a general framework for learning ranking policies under fairness-of-exposure constraints. What remains to be shown is that there exists a stochastic policy class $\Pi$ and

an associated training algorithm that can solve the objective in Equation (1) under the disparities $\mathcal{D}$ defined above. To this effect, we now present the FAIR-PG-RANK algorithm. In particular, we first define a class of Plackett-Luce ranking policies that incorporate a machine learning model, and then present a policy-gradient approach to efficiently optimize the training objective.

## 3.1 Plackett-Luce Ranking Policies

The ranking policies $\pi$ we define in the following comprise of two components: a scoring model that defines a distribution over rankings, and its associated sampling method. Starting with the scoring model $h_\theta$, we allow any differentiable machine learning model with parameters $\theta$, for example a linear model or a neural network. Given an input $\mathbf{x}^q$ representing the feature vectors of all query-document pairs of the candidate set, the scoring model outputs a vector of scores $h_\theta(\mathbf{x}^q) = (h_\theta(x_1^q), h_\theta(x_2^q), \dots h_\theta(x_{n_q}^q))$. Based on this score vector, the probability $\pi_\theta(r|q)$ of a ranking $r = \langle r(1), r(2), \dots r(n_q) \rangle$ under the Plackett-Luce model [29] is the following product of softmax distributions

$$\pi_\theta(r|q) = \prod_{i=1}^{n_q} \frac{\exp(h_\theta(x_{r(i)}^q))}{\exp(h_\theta(x_{r(i)}^q)) + \dots + \exp(h_\theta(x_{r(n_q)}^q))}. \tag{5}$$

Note that this probability of a ranking can be computed efficiently, and that the derivative of $\pi_\theta(r|q)$ and $\log \pi_\theta(r|q)$ exists whenever the scoring model $h_\theta$ is differentiable. Sampling a ranking under the Plackett-Luce model is efficient as well. To sample a ranking, starting from the top, documents are drawn recursively from the probability distribution resulting from Softmax over the scores of the remaining documents in the candidate set, until the set is empty.

## 3.2 Policy-Gradient Training Algorithm

The next step is to search this policy space $\Pi$ for a model that maximizes the objective in Equation (1). This section proposes a policy-gradient approach [30, 31], where we use stochastic gradient descent (SGD) updates to iteratively improve our ranking policy. However, since both $U$ and $\mathcal{D}$ are expectations over rankings sampled from $\pi$, computing the gradient brute-force is intractable. In this section, we derive the required gradients over expectations as an expectation over gradients. We then estimate this expectation as an average over a finite sample of rankings from the policy to get an approximate gradient.

Conventional LTR methods that maximize user utility are either designed to optimize over a smoothed version of a specific utility metric, such as SVMRank [32], RankNet [33] etc., or use heuristics to optimize over probabilistic formulations of rankings (e.g. SoftRank [34]). Our LTR setup is similar to ListNet [35], however, instead of using a heuristic loss function for utility, we present a policy gradient method to directly optimize over both utility and disparity measures. Directly optimizing the ranking policy via policy-gradient learning has two advantages over most conventional LTR algorithms, which optimize upper bounds or heuristic proxy measures. First, our learning algorithm directly optimizes a specified user utility metric and has no restrictions in the choice of the information retrieval (IR) metric. Second, we can use the same policy-gradient approach on our disparity measure $\mathcal{D}$ as well, since it is also an expectation over rankings. Overall, the use of policy-gradient optimization in the space of stochastic ranking policies elegantly handles the non-smoothness inherent in rankings.

### 3.2.1 PG-RANK: Maximizing User Utility

The user utility of a policy $\pi_\theta$ for a query $q$ is defined as $U(\pi|q) = \mathbb{E}_{r \sim \pi_\theta(r|q)} \Delta(r, \mathrm{rel}^q)$. Note that taking the gradient w.r.t. $\theta$ over this expectation is not straightforward, since the space of rankings is exponential in cardinality. To overcome this, we use sampling via the log-derivative trick pioneered in the REINFORCE algorithm [30] as follows:

$$\nabla_\theta U(\pi_\theta|q) = \nabla_\theta \mathbb{E}_{r \sim \pi_\theta(r|q)} \Delta(r, \mathrm{rel}^q) = \mathbb{E}_{r \sim \pi_\theta(r|q)} [\nabla_\theta \log \underbrace{\pi_\theta(r|q)}_{\text{Eq. (5)}} \Delta(r, \mathrm{rel}^q)] \tag{6}$$

This transformation exploits that the gradient of the expected value of the metric $\Delta$ over rankings sampled from $\pi$ can be expressed as the expectation of the gradient of the log probability of each

sampled ranking multiplied by the metric value of that ranking. The final expectation is approximated via Monte-Carlo sampling from the Plackett-Luce model in Eq. (5).

Note that this policy-gradient approach to LTR, which we call PG-RANK, is novel in itself and beyond fairness. It can be used as a standalone LTR algorithm for virtually any choice of utility metric $\Delta$, including NDCG, DCG, ERR, and Average-Rank. Furthermore, PG-RANK also supports non-linear metrics, IPS-weighted metrics for partial information feedback [26], and listwise metrics that do not decompose as a sum over individual documents [36].

**Using baseline for variance reduction.** Since making stochastic gradient descent updates with this gradient estimate is prone to high variance, we subtract a baseline term from the reward [30] to act as a control variate for variance reduction. Specifically, in the gradient estimate in Eq. (6), we replace $\Delta(r, \mathrm{rel}^q)$ with $\Delta(r, \mathrm{rel}^q) - b(q)$ where $b(q)$ is the average $\Delta$ for the current query.

**Entropy Regularization** While optimizing over stochastic policies, entropy regularization is used as a method for encouraging exploration as to avoid convergence to suboptimal deterministic policies [37, 38]. For our algorithm, we add the entropy of the probability distribution $\mathrm{Softmax}(h_\theta(\mathbf{x}^q))$ times a regularization coefficient $\gamma$ to the objective.

### 3.2.2 Minimizing disparity

When a fairness-of-exposure term $\mathcal{D}$ is included in the training objective, we also need to compute the gradient of this term. Fortunately, it has a structure similar to the utility term, so that the same Monte-Carlo approach applies. Specifically, for the **individual-fairness disparity** measure in Equation (3), the gradient can be computed as:

$$\nabla_\theta \mathcal{D}_{\mathrm{ind}} = \frac{1}{|H|} \sum_{(i,j) \in H} \mathbb{1}\left[\left(\frac{v_\pi(d_i)}{M_i} - \frac{v_\pi(d_j)}{M_j}\right) > 0\right] \times \mathbb{E}_{r \sim \pi_\theta(r|q)}\left[\left(\frac{v_r(d_i)}{M_i} - \frac{v_r(d_j)}{M_j}\right)\nabla_\theta \log \pi_\theta(r|q)\right]$$

$$(H = \{(i,j) \text{ s.t. } M_i \geq M_j\})$$

For the **group-fairness disparity** measure defined in Equation (4), the gradient can be derived as follows:

$$\nabla_\theta \mathcal{D}_{\mathrm{group}}(\pi|G_0, G_1, q) = \nabla_\theta \max\left(0, \xi_q \mathrm{diff}(\pi|q)\right) = \mathbb{1}\left[\xi_q \mathrm{diff}(\pi|q) > 0\right] \xi_q \nabla_\theta \mathrm{diff}(\pi|q)$$

where $\mathrm{diff}(\pi|q) = \left(\frac{v_\pi(G_0)}{M_{G_0}} - \frac{v_\pi(G_1)}{M_{G_1}}\right)$, and $\xi_q = \mathrm{sign}(M_{G_0} - M_{G_1})$.

$$\nabla_\theta \mathrm{diff}(\pi|q) = \mathbb{E}_{r \sim \pi_\theta}\left[\left(\frac{\sum_{d \in G_0} v_r(d)}{\sum_{d \in G_0} M(\mathrm{rel}_d)} - \frac{\sum_{d \in G_1} v_r(d)}{\sum_{d \in G_1} M(\mathrm{rel}_d)}\right)\nabla_\theta \log \pi_\theta(r|q)\right]$$

The derivation of the gradients is shown in the supplementary material. The expectation of the gradient in both the cases can be estimated as an average over a Monte Carlo sample of rankings from the distribution. The size of the sample is denoted by $S$ in the rest of the paper.

The completes all necessary ingredients for SGD training of objective (1), and now we present all steps of the FAIR-PG-RANK algorithm.

### 3.3 Summary of the FAIR-PG-RANK algorithm

Algorithm 1 summarizes our method for learning fair ranking policies given a training dataset.

## 4 Empirical Evaluation

We conduct experiments on simulated and real-world datasets to empirically evaluate our approach. First, in Section § 4.1, we validate that the policy-gradient algorithm is competitive with conventional LTR approaches independent of fairness considerations. We accomplish this by comparing our method PG-RANK relative to conventional LTR baselines on the Yahoo! Learning-to-Rank dataset. Second, in Section § 4.2, we use simulated data to verify that FAIR-PG-RANK can detect and mitigate unfair features. Third, we show the effectiveness of our algorithm on real-world datasets by presenting experiments on the Yahoo! Learning to Rank dataset for individual fairness and the German Credit Dataset [39] for group fairness (Section § 4.3).

**Algorithm 1** FAIR-PG-RANK

---

**Input:** $\mathcal{T} = \{(\mathbf{x}^q, \text{rel}^q)\}_{i=1}^N$, disparity measure $\mathcal{D}$, utility/fairness trade-off $\lambda$
Parameters: model $h_\theta$, learning rate $\eta$, entropy reg $\gamma$
Initialize $h_\theta$ with parameters $\theta_0$
**repeat**
   $q = (\mathbf{x}^q, \text{rel}^q) \sim \mathcal{T}$ {Draw a query from training set}
   $h_\theta(\mathbf{x}^q) = (h_\theta(x_1^q), h_\theta(x_2^q), \ldots h_\theta(x_{n_q}^q))$ {Obtain scores for each document}
   **for** $i = 1$ **to** $S$ **do**
     $r_i \sim \pi_\theta(r|q)$ {Plackett-Luce sampling}
   **end for**
   $\nabla \leftarrow \hat{\nabla}_\theta U - \lambda \hat{\nabla}_\theta \mathcal{D}$ {Compute gradient as an average over all $r_i$ using § 3.2.1 and § 3.2.2}
   $\theta \leftarrow \theta + \eta \nabla$ {Update}
**until** convergence on the validation set

---

Table 1: Comparing PG-RANK to the baseline LTR methods from [24] on the Yahoo dataset.

|  | NDCG@10 | ERR |
|---|---|---|
| RankSVM [40] | 0.75924 | 0.43680 |
| GBDT [41] | 0.79013 | 0.46201 |
| PG-RANK (Linear model) | 0.76145 | 0.44988 |
| PG-RANK (Neural Network) | 0.77082 | 0.45440 |

For all the experiments, we use NDCG as the utility metric, define merit using the identity function $M(\text{rel}) = \text{rel}$, and set the position bias $\mathbf{v}$ to follow the same distribution as the gain factor in DCG i.e. $\mathbf{v}_j \propto \frac{1}{log_2(1+j)}$ where $j = 1, 2, 3, \ldots$ is a position in the ranking.

### 4.1 Can PG-RANK learn accurate ranking policies?

To validate that PG-RANK is indeed a highly effective LTR method, we conduct experiments on the Yahoo dataset [24]. We use the standard experiment setup on the SET 1 dataset and optimize NDCG using PG-RANK, which is equivalent to finding the optimal policy in Eq. (1) with $\lambda = 0$.

We train FAIR-PG-RANK for two kinds of scoring models: a linear model and a neural network (one hidden layer with 32 hidden units and ReLU activation). Details of the models and training hyperparameters are given in the supplementary material. The policy learned by our method is a stochastic policy, however, for the purpose of evaluation in this task, we use the highest probability ranking of the candidate set for each query to compute the average NDCG@10 and ERR (Expected Reciprocal Rank) over all the test set queries. We compare our evaluation scores with two baselines from Chapelle and Chang [24] – a linear RankSVM [40] and a non-linear regression-based ranker that uses Gradient-boosted Decision Trees (GBDT) [41].

Table 1 shows that PG-RANK achieves competitive performance compared to the conventional LTR methods. When comparing PG-RANK to RankSVM for linear models, our method outperforms RankSVM in terms of both NDCG@10 and ERR. This verifies that the policy-gradient approach is effective at optimizing utility without having to rely on a possibly lose convex upper bound like RankSVM. PG-RANK with the non-linear neural network model further improves on the linear model. Furthermore, additional parameter tuning and variance-control techniques from policy optimization are likely to further boost the performance of PG-RANK, but are outside the scope of this paper.

### 4.2 Can FAIR-PG-RANK effectively trade-off between utility and fairness?

We designed a synthetic dataset to allow inspection into how FAIR-PG-RANK trades-off between user utility and fairness of exposure. The dataset contains 100 queries with 10 candidate documents each. In expectation, 8 of those documents belong to the majority group $G_0$ and 2 belong to the minority group $G_1$. For each document we independently and uniformly draw two values $x_1$ and $x_2$ from the interval $(0, 3)$, and set the relevance of the document to $x_1 + x_2$ clipped between 0 and 5. For the documents from the majority group $G_0$, the features vector $(x_1, x_2)$ representing the documents provides perfect information about relevance. For documents in the minority group $G_1$,

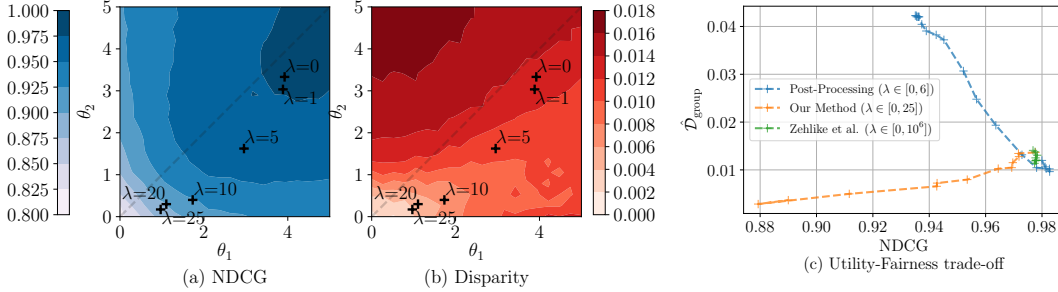

(a) NDCG           (b) Disparity          (c) Utility-Fairness trade-off

Figure 1: Experiments on Simulated dataset. The shaded regions show different ranges of the values of (a) NDCG, (b) Group Disparity ($\mathcal{D}_{\text{group}}$), with varying model parameters $\theta = (\theta_1, \theta_2)$. The (**+**) points show the models learned by FAIR-PG-RANK under different values of $\lambda$. (c) Comparison of NDCG and Group Disparity ($\mathcal{D}_{\text{group}}$) trade-off for different methods.

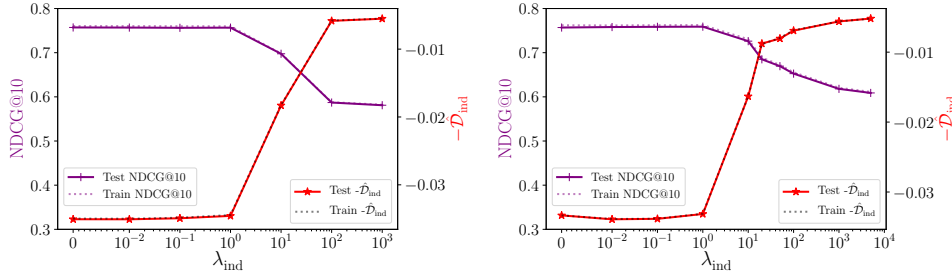

Figure 2: Effect of varying $\lambda$ on NDCG@10 (user utility) and $\mathcal{D}_{\text{ind}}$ (individual fairness disparity) on Yahoo data. *Left*: Linear model, *Right*: Neural Network. The overlapping dotted curves represent the training set NDCG@10 and Disparity, while solid curves show test set performance.

however, feature $x_2$ is corrupted by replacing it with zero so that the information about relevance for documents in $G_1$ only comes from $x_1$. This leads to a biased representation between groups, and any use of $x_2$ is prone to producing unfair exposure between groups.

In order to validate that FAIR-PG-RANK can detect and neutralize this biased feature, we consider a linear scoring model $h_\theta(\mathbf{x}) = \theta_1 x_1 + \theta_2 x_2$ with parameters $\theta = (\theta_1, \theta_2)$. Figure 1 shows the contour plots of NDCG and $\mathcal{D}_{\text{group}}$ evaluated for different values of $\theta$. Note that not only the direction of the $\theta$ vector affects both NDCG and $\mathcal{D}_{\text{group}}$, but also its length as it determines the amount of stochasticity in $\pi_\theta$. The true relevance model lies on the $\theta_1 = \theta_2$ line (dotted), however, a fair model is expected to ignore the biased feature $x_2$. We use PG-RANK to train this linear model to maximize NDCG and minimize $\mathcal{D}_{\text{group}}$. The dots in Figure 1 denote the models learned by FAIR-PG-RANK for different values of $\lambda$. For small values of $\lambda$, FAIR-PG-RANK puts more emphasis on NDCG and thus learns parameter vectors along the $\theta_1 = \theta_2$ direction. As we increase emphasis on group fairness disparity $\mathcal{D}_{\text{group}}$ by increasing $\lambda$, the policies learned by FAIR-PG-RANK become more stochastic and it correctly starts to discount the biased attribute by learning models where increasingly $\theta_1 >> \theta_2$.

In Figure 1(c), we compare FAIR-PG-RANK with two baselines. As the first baseline, we estimate relevances with a fairness-oblivious linear regression and then use the post-processing method from [3] on the estimates. Unlike FAIR-PG-RANK, which reduces disparity with increasing $\lambda$, the post-processing method is mislead by the estimated relevances that use the biased feature $x_2$, and the ranking policies become even less fair as $\lambda$ is increased. As the second baseline, we apply the method of Zehlike and Castillo [23], but the heuristic measure it optimizes shows little effect on disparity.

## 4.3 Can FAIR-PG-RANK learn fair ranking policies on real-world data?

In order to study FAIR-PG-RANK on real-world data, we conducted two sets of experiments.

For **Individual Fairness**, we train FAIR-PG-RANK with a linear and a neural network model on the Yahoo! Learning to rank challenge dataset, optimizing Equation 1 with different values of $\lambda$. The details about the model and training hyperparameters are present in the supplementary material. For both the models, Figure 2 shows the average NDCG@10 and $\mathcal{D}_{\text{ind}}$ (individual disparity) over the test and training (dotted line) datasets for different values of $\lambda$ parameter. As desired, FAIR-PG-RANK emphasizes lower disparity over higher NDCG as the value of $\lambda$ increases, with disparity going down

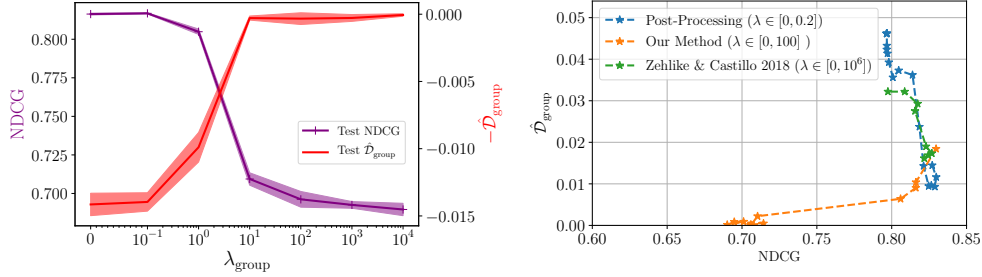

Figure 3: *Left*: Effect of varying $\lambda$ on the test set NDCG and $\mathcal{D}_{\text{group}}$ for the German Credit Dataset. The shaded area shows the standard deviation over five runs of the algorithm on the data. *Right*: Comparison of NDCG and Group Disparity ($\mathcal{D}_{\text{group}}$) trade-off for different methods.

to zero eventually. Furthermore, the training and test curves for both NDCG and disparity overlap indicating the learning method generalizes to unseen queries. This is expected since both training quantities concentrate around their expectation as the training set size increases.

For **Group fairness**, we adapt the German Credit Dataset from the UCI repository [39] to a learning-to-rank task (described in the supplementary), choosing gender as the group attribute. We train FAIR-PG-RANK using a linear model, for different values of $\lambda$. Figure 3 shows that FAIR-PG-RANK is again able to effectively trade-off NDCG and fairness. Here we also plot the standard deviation to illustrate that the algorithm reliably converges to solutions of similar performance over multiple runs. Similar to the synthetic example, Figure 3 (*right*) again shows that FAIR-PG-RANK can effectively trade-off NDCG for $\mathcal{D}_{\text{group}}$, while the baselines fail.

## 5 Conclusion

We presented a framework for learning ranking functions that not only maximize utility to their users, but that also obey application specific fairness constraints on how exposure is allocated to the ranked items based on their merit. Based on this framework, we derived the FAIR-PG-RANK policy-gradient algorithm that directly optimizes both utility and fairness without having to resort to upper bounds or heuristic surrogate measures. We demonstrated that our policy-gradient approach is effective for training high-quality ranking functions, that FAIR-PG-RANK can identify and neutralize biased features, and that it can effectively learn ranking functions under both individual fairness and group fairness constraints.

## Acknowledgements

This work was supported in part by a gift from Workday Inc., as well as NSF awards IIS-1615706, IIS-1513692, and IIS-1901168. We thank Jessica Hong for the interesting discussions that informed the direction of this paper. Any opinions, findings, and conclusions or recommendations expressed in this material are those of the author(s) and do not necessarily reflect the views of the National Science Foundation.

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
