[Supplementary Material]

# Supplementary Material: Policy Learning for Fairness in Ranking

**Ashudeep Singh**
Department of Computer Science
Cornell University
Ithaca, NY 14850
ashudeep@cs.cornell.edu

**Thorsten Joachims**
Department of Computer Science
Cornell University
Ithaca, NY 14850
tj@cs.cornell.edu

## A  Policy Gradient for PL Ranking policy

In this section, we will show the derivation of gradients for utility $U$ and disparity ($\mathcal{D}_{\text{group}}$ and $\mathcal{D}_{\text{ind}}$). Since both $U$ and $\mathcal{D}$ are expectations over rankings sampled from $\pi$, computing the gradient brute-force is intractable. We derive the required gradients over expectations as an expectation over gradients. We then estimate this expectation as an average over a finite sample of rankings from the policy to get an approximate gradient. Later, we also present a summary of the FAIR-PG-RANK algorithm.

### A.1  Gradient of Utility measures

To overcome taking a gradient over expectations, we use the log-derivative trick pioneered in the REINFORCE algorithm (Williams, 1992) as follows

$$
\begin{aligned}
\nabla_\theta U(\pi_\theta|q) &= \nabla_\theta \mathbb{E}_{r \sim \pi_\theta(r|q)} \Delta\big(r, \text{rel}^q\big) \\
&= \nabla_\theta \sum_{r \in \sigma(n_q)} \pi_\theta(r|q) \Delta\big(r, \text{rel}^q\big) \\
&= \sum_{r \in \sigma(n_q)} \nabla_\theta[\pi_\theta(r|q)] \Delta\big(r, \text{rel}^q\big) \\
&= \sum_{r \in \sigma(n_q)} \pi_\theta(r|q) \nabla_\theta[\log \pi_\theta(r|q)] \Delta\big(r, \text{rel}^q\big) \qquad \text{(Log-derivative trick [1])} \\
&= \mathbb{E}_{r \sim \pi_\theta(r|q)}[\nabla_\theta \log \pi_\theta(r|q) \Delta(r, \text{rel}^q)]
\end{aligned}
$$

The expectation over $r \sim \pi_\theta(r|q)$ can be computed as an average over a finite sample of rankings from the policy.

### A.2  Gradient of Disparity functions

The gradient of the disparity measure for individual fairness can be derived as follows:

$$
\begin{aligned}
\nabla_\theta \mathcal{D}_{\text{ind}} &= \nabla_\theta \left[ \frac{1}{|H|} \sum_{(i,j) \in H} \max\left(0, \frac{v_\pi(d_i)}{M_i} - \frac{v_\pi(d_j)}{M_j}\right) \right] \qquad (H = \{(i,j) \text{ s.t. } M_i \geq M_j\}) \\
&= \nabla_\theta \left[ \frac{1}{|H|} \sum_{(i,j) \in H} \max\big(0, \text{pdiff}_q(\pi, i, j)\big) \right] \\
&= \frac{1}{|H|} \sum_{(i,j) \in H} \mathbb{1}[\text{pdiff}_q(\pi, i, j) > 0] \nabla_\theta \text{pdiff}_q(\pi, i, j)
\end{aligned}
$$

$$\nabla_\theta \text{pdiff}_q(\pi, i, j) = \nabla_\theta \left[ \frac{v_\pi(d_i)}{M_i} - \frac{v_\pi(d_j)}{M_j} \right]$$

$$= \nabla_\theta \mathbb{E}_{r \sim \pi_\theta(r|q)} \left[ \frac{v_r(d_i)}{M_i} - \frac{v_r(d_i)}{M_j} \right]$$

$$= \mathbb{E}_{r \sim \pi_\theta(r|q)} \left[ \left( \frac{v_r(d_i)}{M_i} - \frac{v_r(d_i)}{M_j} \right) \nabla_\theta \log \pi_\theta(r|q) \right] \qquad \text{(using the log-derivative trick)}$$

The gradient of the disparity measure for group fairness can be derived as follows:

$$\nabla_\theta \mathcal{D}_{\text{group}}(\pi|G_0, G_1, q) = \nabla_\theta \max\big(0, \xi_q \text{diff}(\pi|q)\big)$$

where $\text{diff}(\pi|q) = \left( \frac{v_\pi(G_0)}{M_{G_0}} - \frac{v_\pi(G_1)}{M_{G_1}} \right)$, and $\xi_q = +1$ if $M_{G_0} \geq M_{G_1}, \; -1$ otherwise. Further,

$$\nabla_\theta \mathcal{D}_{\text{group}}(\pi|G_0, G_1, q) = \mathbb{1}\big[ \xi_q \text{diff}(\pi|q) > 0 \big] \xi_q \nabla_\theta \text{diff}(\pi|q)$$

$$\text{where, } \nabla_\theta \text{diff}(\pi_\theta|q) = \nabla_\theta \left[ \frac{v_\pi(G_0)}{M_{G_0}} - \frac{v_\pi(G_1)}{M_{G_1}} \right]$$

$$= \nabla_\theta \left[ \frac{\frac{1}{|G_0|} \sum_{d \in G_0} \mathbb{E}_{r \sim \pi_\theta} v_r(d)}{\frac{1}{|G_0|} \sum_{d \in G_0} M(\text{rel}_d)} - \frac{\frac{1}{|G_1|} \sum_{d \in G_1} \mathbb{E}_{r \sim \pi_\theta} v_r(d)}{\frac{1}{|G_1|} \sum_{d \in G_1} M(\text{rel}_d)} \right]$$

$$= \nabla_\theta \mathbb{E}_{r \sim \pi_\theta} \left[ \frac{\sum_{d \in G_0} v_r(d)}{\sum_{d \in G_0} M(\text{rel}_d)} - \frac{\sum_{d \in G_1} v_r(d)}{\sum_{d \in G_1} M(\text{rel}_d)} \right]$$

$$= \mathbb{E}_{r \sim \pi_\theta} \left[ \left( \frac{\sum_{d \in G_0} v_r(d)}{\sum_{d \in G_0} M(\text{rel}_d)} - \frac{\sum_{d \in G_1} v_r(d)}{\sum_{d \in G_1} M(\text{rel}_d)} \right) \nabla_\theta \log \pi_\theta(r|q) \right]$$

Similarly, the expectation over $r \sim \pi_\theta(r|q)$ can be computed as an average over a finite sample of rankings from the policy.

## B  Datasets and Models

### B.1  Yahoo! Learning to Rank dataset

We used SET 1 from the Yahoo! Learning to Rank challenge [2], which consists of $19,944$ training queries and $6,983$ queries in the test set. Each query has a variable sized candidate set of documents that needs to be ranked. There are a total of $473,134$ and $165,660$ documents in training and test set respectively. The query-document pairs are represented by a 700-dimensional feature vector. For supervision, each query-document pair is assigned an integer relevance judgments from 0 (bad) to 4 (perfect).

### B.2  German Credit Dataset

The original German Credit dataset [3] consists of 1000 individuals, each described by a feature vector $x_i$ consisting of 20 attributes with both numerical and categorical features, as well as a label $\text{rel}_i$ classifying it as creditworthy ($\text{rel}_i = 1$) or not ($\text{rel}_i = 0$). We adapt this binary classification task to a learning-to-rank task in the following way: for each query q, we sample a candidate set of 10 individuals each, randomly sampling irrelevant documents (non-creditworthy individuals) and relevant documents (creditworthy individuals) in the ratio 4:1. Each individual is identified as a member of group $G_0$ or $G_1$ based on their gender attribute.

### B.3  Baselines

We compare our method to two methods:

1. Post-processing method on estimated relevances: First, we train a linear regression model on all the training set query-document pairs that predicts their relevances. For each query in the test set, we use the estimated relevances of the documents as an input to the linear

program from Singh and Joachims [4] with the disparate exposure constraint for group fairness (section § 2.2). We use the following linear program to find the optimal ranking that satisfies fairness constraints on estimated relevances:

$$\mathbb{P}^* = \text{argmax}_{\mathbb{P}} \quad \mathbf{u}^T \mathbb{P} \mathbf{v} - \lambda \xi \quad \text{(where } \mathbf{u}_i = 2^{\hat{\text{rel}}_i} - 1 \text{ and } \mathbf{v}_j = \frac{1}{\log 1+j} \text{ as in § 2.1)}$$

$$\text{s.t.} \quad \forall j \quad \sum_j \mathbb{P}_{ij} = 1 \qquad \text{(sum of probabilities for each document)}$$

$$\forall i \quad \sum_i \mathbb{P}_{ij} = 1 \qquad \text{(sum of probabilities at each position)}$$

$$\forall i,j \quad 0 \leq \mathbb{P}_{ij} \leq 1 \qquad \text{(valid probabilities)}$$

$$M(G_k) \geq M(G_{k'}) \Rightarrow \left( \frac{\sum_{d_i \in G_k} \mathbb{P}_i^T \mathbf{v}}{M(G_k)} - \frac{\sum_{d_i \in G_{k'}} \mathbb{P}_i^T \mathbf{v}}{M(G_{k'})} \right) \geq -\xi$$
$$\text{(Disparate exposure fairness constraint)}$$

$$\xi \geq 0$$

Note that the relevances used in the linear program (in $\mathbf{u}$) are estimated relevances. This is one of the reasons that even when using this linear program to minimize disparity, we cannot guarantee that disparity on unseen queries can be reduced to zero. In contrast to Singh and Joachims [4], rather than solving the exact constraint, we use a $\lambda$ hyperparameter to control how much unfairness we can allow. For our experiments, we evaluate the performance for values of $\lambda \in [0, 0.2]$ (at $\lambda = 0.2$, for all queries the disparity measure on estimated relevances was reduced to zero).

The linear program outputs a $n_q \times n_q$-sized probabilistic matrix $\mathbb{P}$ representing the probability of each document at each position. We compare the NDCG and $\mathcal{D}_{\text{group}}$ for this probabilistic matrix to other methods in sections § 4.2 and § 4.3.

2. Zehlike and Castillo [5]: This method uses a cross-entropy loss on the top-1 probability of each document to maximize utility. The top-1 probabilities of each document is obtained through a Softmax over scores output by a linear scoring function. The disparity measure is implemented as the squared loss of the difference between the top-1 exposure of the groups $G_0$ and $G_1$. Training is done using stochastic gradient descent on the sum of cross entropy and $\lambda$ times the disparity measure. For all our experiments with this method, we didn't use any regularization, searched for the best learning rate in the range $[10^{-3}, 1]$, and evaluated the performance for $\lambda \in \{0, 1, 10, 10^2, \ldots, 10^6\}$.

### B.4 Model and Training: Yahoo! Learning to Rank challenge dataset

We train two different models for experiments in Section § 4.1: a linear model, and a neural network. The neural network has one hidden layer of size 32 and ReLU activation function. For training, all the weights were randomly initialized between $(-0.001, 0.001)$ for the linear model and $(-1/\sqrt{32}, 1/\sqrt{32})$ for the neural network. We use an Adam optimizer with a learning rate of 0.001 for the linear model and $5 \times 10^{-5}$ for the neural network. For both the cases, we set the entropy regularization constant to $\gamma = 1.0$, use a baseline, and use a sample size of $S = 10$ to estimate the gradient. Both models are trained for 20 epochs over the training dataset, updating the model one query at a time.

### B.5 Model and Training: German Credit Dataset

To validate whether FAIR-PG-RANK can also optimize for Group fairness, we used the modified German Credit Dataset from the UCI repository (section § B.2). We train a linear scoring model with Adam, using a fixed learning rate of 0.001 with no regularization, and a sample size $S = 25$, for different values of $\lambda$ in the range $[0, 25]$. We compare our method to baselines mentioned in § B.3.