[Reviews · NeurIPS 2019]

Reviewer 1



The paper is fairly interesting in terms of motivation, but the actual execution perhaps isn't mature enough the way it has been presented. -- since this is a new fairness criterion, a more detailed discussion of its merits would be helpful (even if it is non-rigorous). in particular, it is unclear why the disparity is only measured in one direction (over-emphasis of higher relevance item) with the direction based on relevance rather than group identity? Ideally, fairness critera would be defined in a way that is cognisant of historical/natural directions of bias, and therefore checks over/under-emphasis based on group identity rather than utility (which is dealt with separately). given that, this paper is more of a "diversity" metric rather than a "fairness" metric -- the experimental results are a bit confusing, not least because some of the axes measure D and others measure -D. In particular, figure 3 (right) seems to suggest that the post-processing method has a configuration where increasing the NDCG score also decreases the disparity? Isn't that a good thing? -- the kinds of experiments presented are all over the place-- the yahoo dataset and the german credit dataset show entirely different types of experiments, which makes it difficult to assess -- in figure 2, the dashed and the solid lines (train and test results respectively) seem suspiciously close to each other. but since the authors have provided code (that i did not check) i am willing to give them the benefit of the doubt on this one. -- the simulated data experiments, while mildly interesting, is too much of a toy experiment to take too seriously otherwise, the paper is fairly clear and reasonably well-motivated and includes most of the relevant references. this can be a pretty good paper with some improvement, but i don't think it is there just yet.

Reviewer 2



a new algorithm provided for ranking named as Fair-PG-Rank which seems to provide useful results.

Reviewer 3



This paper presents a framework for expressing specifications for ranking fairness, along with a new learning to rank algorithm based on policy-gradient approach, which can support various fairness constraints. Overall, it is extremely well written, and makes important contributions in an area of rapidly growing importance. Instead of starting with an existing LTR algorithm, this paper takes a fresh view on LTR as the problem of learning a stochastic ranking policy, which is learnt via ERM. This is the most important and novel contribution of the paper. Next, they propose a class of fairness constraints for ranking that incorporate both individual and group fairness, building on previous related work to adapt to the learning procedure. Finally, they show a policy gradient approach for directly optimizing *any* IR utility metric trading off with a variety of fairness criteria. They show the effectiveness of their proposed approach on both synthetic and real world datasets. Ironically, the motivating example presented to explain the tradeoff of utility and exposure, though only illustrative, is slightly biased itself. My suggestion would be to reverse the genders in the example to avoid perpetuating implicit biases about women’s merit. One concern in the ERM formulation is that NDCG is typically used as NDCG@k. I believe this should be handled by appropriately setting the position bias values, but not sure if it would introduce some discontinuities in the optimization problem. Some clarification on this would be useful. Another confusing part about the formulation is the handling of the parameter \delta, which is the maximum expected disparity. I would think that it would be more desirable for the designer for the model to be able to specify the value of \delta based on the domain requirement, and then we should minimize w.r.t. \lambda for a chosen \delta. The idea of letting the model designer steer the utility/fairness trade-off might be useful in certain settings, but not in others. This paper makes two very interesting improvements over the previous work in fairness constraints: one is the proposal of constraints that enforce the proportionality of exposure to merit. The idea that “higher merit items don’t get exposure beyond their merit, since the opposite direction is already achieved through utility maximization” is a pretty powerful one. Second, the proposed measures of disparate exposure for individual and group fairness can be very useful by themselves as tests of fairness for any given ranking system. One clarification that would be nice is the use of “merit” as a function of “relevance”, which is not well motivated. It’s not clear why we need this, instead of directly using relevance, which is indeed used in the experiments. The FAIR-PG-RANK algorithm is well designed, but it does have the limitation of allowing only differential ML models, which does exclude some popular LTR approaches. The log derivative trick is extremely clever, and does open up a lot of possibilities. Again, some elaboration of what happens in the NDCG@K case would be extremely useful. Given the overall technical strength of the paper, the empirical evaluation does leave much to be desired. For example, the proposed LTR approach is only compared with very few, and relatively older methods. Even though the idea is to just show that this approach is competitive, comparisons to more recent and widely used algorithms like LambdaRank, LambdaMART or their successors would have been better. Also, the argument that it does worse than GBDT since it’s different model class if pretty weak. The synthetic data experiments could also have been repeated on larger document sets for better understanding of model behavior. It’s not clear why group parity is not studied using synthetic data as well. Finally, the experiments on German Credit Dataset demonstrate the generalization and trade off properties, but some comparison with some modified version of other fairness approaches, such as top-K, or one of the supervised classification ones would make this paper much stronger. Overall, the paper is well written, and makes strong contributions to the fairness in ranking field. The empirical evaluation is somewhat weak, but there are enough high impact ideas proposed here for it to be accepted.

[Author Response · NeurIPS 2019]

We thank all reviewers for their valuable and helpful comments. They are addressed in detail in our response below.

**Reviewer # 1**:

**"... it is unclear why the disparity is only measured in one direction (over-emphasis of higher relevance item) with the direction based on relevance ..."** We will expand the current explanation (Line 137-144 in Section 2.2) in the final version, further elaborating why we designed the metric to be one-sided. Most importantly, imposing the two-sided proportionality constraint in Singh & Joachims (2018) and Biega et al. (2018) results in an infeasible problem when a small portion of groups/items contribute most of the relevance for the query. Consider the extreme case where only one item is relevant. After putting this item at rank one, we have to put some non-relevant item in position two. This item is now overexposed and violates the two-sided metric, but not the one-sided metric. In this way, the one-sided metric together with utility maximization allows non-relevant items to get higher exposure when this is unavoidable.

**"Ideally, fairness critera would be defined in a way that is cognisant of historical/natural directions of bias ..."** We appreciate your idea of selecting the direction of the one-sided metric based on historical disadvantage. We will further clarify the generality of our algorithm in the final version — specifically that such constraints, driven by specific application concerns, can easily be trained with the proposed policy-gradient algorithm as well. We focused on the metrics in Equations (3) and (4) for their broad applicability. For example, when ensuring fair exposure of products (items) by different sellers (groups) in a marketplace, there might not be a no notion of historical disadvantage.

**"... figure 3 (right) seems to suggest that the post-processing method has a configuration where increasing the NDCG score also decreases the disparity?"** There is indeed a tiny increase in NDCG in the second and third point. Note that if the post-processing method had access to the ground-truth relevances, this increase wouldn't happen and both NDCG and fairness would change monotonically with $\lambda$. However, the post-processing method has to work with regression estimates of item relevances, that are inaccurate and possibly biased. This leads to the erratic behavior of the post-processing method, and the method actually decreases fairness for much of the range of $\lambda$ making it impractical, while our policy-gradient method allows a meaningful selection of $\lambda$ that succeeds in driving unfairness to zero.

**"the kinds of experiments presented are all over the place– the yahoo dataset and the german credit dataset show entirely different types of experiments, which makes it difficult to assess".** Thank you for the feedback. To clarify, the experiments were designed with three very specific goals in mind:
(Section 4.1) How does our method perform relative to traditional LTR approaches when not using fairness constraints? To be comparable to the state of the art, we used the standard Yahoo! LTR challenge dataset without any modifications. This experiment verifies that our policy-gradient approach achieves reasonable ranking performance.
(Section 4.2) Does the fair ranking model learn to ignore the biased feature? We chose a synthetic dataset that can be visualized to provide direct evidence for our claim that the method can learn to discount biased features. This is a key property that sets it apart from post-processing methods where the learning step is indifferent to fairness, making it impossible for post-processing methods to recover from a biased regression estimates.
(Section 4.3) Can our method effectively enforce both individual and group fairness constraints on real datasets? To evaluate the ability to enforce individual fairness constraints, we again use the Yahoo! LTR dataset for consistency with Section 4.1 and easy reproducability. Unfortunately, the Yahoo! LTR dataset does not provide suitable attributes to define groups for evaluating group fairness. We therefore adapted the German Credit dataset to evaluate the ability to enforce group fairness.
For the final version, we will clarify the layout of the experiments section currently presented in lines 238-243 and re-align the figures with text description.

**Reviewer # 3**

**"Ironically, the motivating example ... is slightly biased itself."** Thank you for the thoughtful feedback. It seems that our point about system-endogenous amplification (through disparate exposure) of exogenous biases (e.g. 1% of the employers may be biased against women) is not coming through. We will think about further clarifying, or we will simply replace male/female with neutral group labels.

**"... NDCG is typically used as NDCG@k."** You are right that NDCG@k is just another set of position weights. Specifically, note that the gradient doesn't vanish even for small k for probabilistic policies, although it may be necessary to use a larger sample of rankings for gradient estimation.

**"... minimize w.r.t. $\lambda$ for a chosen $\delta$."** To achieve this behavior, one could simply implement a wrapper that searches for $\lambda$ given $\delta$. While more efficient solutions might exist, such search is common practice in ML (e.g. regularization parameter).

**"... limitation of allowing only differential ML models..."** Yes, our method works only for differentiable models, but this is a pretty large class including deep neural networks, SVMs, Lasso, conditional random fields, matrix factorization, etc.

[Meta-Review · NeurIPS 2019]

All the reviewers agreed that the work tackles an interesting, timely topic and the methodological contribution is sound. As a result, the paper was discussed during the PC meeting and acceptance was recommended. That being said, there were also several concerns that were brought up during the discussion among reviewers and I would encourage the authors to address them in the final version of the paper. In particular: (i) The motivation of the fairness metrics appears insufficient. A more detailed discussion of its merits would be helpful -- it is unclear why the disparity is only measured in one direction (over-emphasis of higher relevance item) with the direction based on relevance rather than group identity. As stated, the metric seems to reward "diversity" rather than "fairness". (ii) The execution of the experimental evaluation could be significantly improved. More specifically, the yahoo dataset and the german credit dataset show entirely different types of experiments, the proposed LTR approach is only compared with very few, and relatively older methods (a comparison to more recent and widely used algorithms like LambdaRank, LambdaMART or their successors seems necessary), and the argument that it does worse than GBDT since it’s different model class if pretty weak. (iii) There are other datasets that may be better suited for validating the authors' method. For example, the ones used in: 1. Zehlike et al. "Fa*ir: A fair top-k ranking algorithm." 2. Asudeh et al. "Designing fair ranking schemes." 3. Biega et al, "Equity of Attention: Amortizing Individual Fairness in Rankings"